# Withanolides, Extracted from Datura Metel L. Inhibit Keratinocyte Proliferation and Imiquimod-Induced Psoriasis-Like Dermatitis via the STAT3/P38/ERK1/2 Pathway

**DOI:** 10.3390/molecules24142596

**Published:** 2019-07-17

**Authors:** Tingting Li, Zheng Wei, Yanping Sun, Qiuhong Wang, Haixue Kuang

**Affiliations:** 1Key Laboratory of Chinese Materia Medica, Heilongjiang University of Chinese Medicine, 24 Heping Road, Xiangfang District, Harbin 150040, China; 2School of Traditional Chinese Medicine, Guangdong Pharmaceutical University, 280 Outer Ring Road, University Town, Guangzhou 510006, China

**Keywords:** psoriasis, withanolides, *Datura metel* L., imiquimod, inflammation

## Abstract

Psoriasis is an immune-mediated inflammatory dermatosis characterized by epidermal hyperplasia and excessive infiltration of inflammatory cells. Withanolides, extracted from *Datura metel* L.; are the main effective components for the treatment of psoriasis. However, the precise mechanisms of action of withanolides for the treatment of psoriasis remain unclear. We found that treatment with withanolides alleviated imiquimod (IMQ)-induced epidermal hyperplasia and inflammatory cell infiltration in the effective skin of model mice. In addition, we also found that withanolides suppressed the activation of STAT3, ERK1/2 and P38 signaling pathways in IMQ-stimulated HaCat cells. These results suggest that withanolides possess an anti-inflammatory effect and have significant therapeutic potential for the prevention and treatment of psoriasis.

## 1. Introduction

Psoriasis is an immune-mediated inflammatory dermatosis characterized by epidermal hyperplasia and excessive infiltration of inflammatory cells [1,2,3,4,5]. The prevalence rate of psoriasis is 2–3% worldwide [6]. Most psoriasis patients also have chronic complications, including arthritis, metabolic syndrome and cardiovascular diseases [7,8]. Multiple complex factors can contribute to the development of psoriasis, such as genetic and environmental factors. Damage to the dermis and epidermis, surgical wounds, bacterial infections, strong sunlight, physiological stress and smoking are other well-known triggers of psoriasis [9]. Pathological changes are associated with psoriasis, including abnormal proliferation and differentiation of epidermal keratinocytes, excessive infiltration of inflammatory cells, such as T cells, dendritic cells (DCs), macrophages and neutrophils and increased skin angiogenesis [10,11]. The activation of T cells, DCs and their upregulation of pro-inflammatory factors are considered to mainly affect the pathogenic development of psoriasis [12]. Pro-inflammatory cytokines IL-6, IL-23, IL-22, IL-17A, IFN-γ and TNF-α have an important role in inflammation, and also affect hyperproliferation and terminal differentiation of keratinocytes [13,14,15].

Flos daturae (Yangjinhua), the dry flowers of *Datura metel* L.; is a traditional Chinese medicine that has been widely used in the treatment of pain, asthma, rheumatism, coughs, and convulsions etc. [16]. *Datura metel* L. belongs to the Solanaceae family. Flos daturae has been officially listed in the Chinese pharmacopoeia. In recent years, accumulating studies have reported that *Datura metel* L. has significant anti-psoriasis effects, and these have been clinically validated at the First Affiliated Hospital of Heilongjiang University of Chinese Medicine [17,18]. It was reported that withanolides extracted from *Datura metel* L. had an inhibitory effect on immune responses [19]. However, the precise therapeutic mechanism(s) of withanolides in psoriasis remains controversial. Withasteroids are a group of structurally diverse steroidal compounds with a C28 steroidal lactone skeleton, in which a characteristic feature is the presence of an α, β-unsaturated δ lactone ring in the side chain. Withasteroids primarily appeared in the Solanaceae family, and they have various biological activities, including anti-tumor, immune-suppressive, anti-inflammatory and chemoprevention properties [20,21].

Imiquimod (IMQ), a Toll-like receptor (TLR7/8) agonist, is applied to mice skin to induce erythema, scaling, acanthosis, parakeratosis, and inflammation. IMQ-induced psoriasis-like mouse model and cell model have been widely used to mimic inflammation-type psoriasis, and these models are of benefit for facilitating the mechanisms of drugs underlying treatment of psoriasis [22,23].

In this study, we aim to evaluate the therapeutic benefits of withanolides using an IMQ-induced psoriasis-like mouse model and human keratinocytes stimulated with IMQ.

## 2. Results

### 2.1. Withanolides Significantly Alleviated IMQ-Induced Psoriasis-Like Symptoms

The morphological observations of back skin after a six-day treatment are shown in Figure 1. The back skin of mice in the control group was normal without any psoriasis-like symptoms. However, the mice in the IMQ-treated group obviously had flaky erythema, severe infiltration and thickened skin. Compared with the IMQ-treated group, withanolides 500 mg/kg group, 250 mg/kg withanolides-treated groups and methotrexate (MTX)-treated group showed less erythema and thin scales and no obvious infiltration; but the improvement of withanolides 1 g/kg group was better than that in withanolides 0.5 g/kg group, similar to MTX-treated group.

We evaluated the severity of erythema, scales, infiltration and total scores of skin lesions on day 1–6 via the Psoriasis Area and Severity Index (PASI) scores. The trend line of PASI scores are shown in Figure 2. The PASI scores of all groups gradually increased with the administration of IMQ. However, PASI scores of withanolides-treated groups exhibited obviously lower than IMQ-treated group. No statistically significant difference was exhibited among the withanolides 500 mg/kg group and the MTX-treated group. The decrease in PASI suggested that withanolides had anti-psoriatic efficacy compared with the IMQ-treated group.

As shown in Figure 3, hematoxylin-eosin (HE)staining showed that the histological changes of the lesions in IMQ-treated group were mainly significant parakeratosis, sudden downward extension and infiltration of inflammation cells into the dermis. Microscopic observations showed that the epidermal thickness of IMQ-treated group was about four times than that in the control group. After six days of treatment with 500 mg/kg and 250 mg/kg withanolides, the treated mice in withanolides 500 mg/kg and 250 mg/kg groups showed less psoriasis-like symptoms, including less parakeratosis, epidermal thickness and infiltration of inflammatory cells compared with IMQ-treated group. The epidermal thickness was significantly decreased, withanolides 500 mg/kg group was similar to the MTX-treated group. In short, the results revealed that intragastric administration of withanolides markedly alleviated psoriasis-like symptoms, indicating that withanolides might be used in the treatment of psoriasis.

### 2.2. Withanolides Inhibited the Proliferation of the Epidermis in Mice with IMQ-Induced Psoriasis

Ki67 is a hallmark of keratinocyte proliferation which is associated with cell cycle. The abnormal expression of Ki67 is the common characteristic of psoriasis [24]. The immunohistochemical staining results showed that the Ki67 positive cells were stained brown in the picture (Figure 4). The IMQ-treated group expressed the most Ki67 compared with other groups. Ki67 positive cells/field of IMQ-treated group accounted for about 8.6% in the picture. There was a statistically significant difference between control group and IMQ-treated group. However, withanolides effectively inhibited the expression of Ki67. Compared to IMQ-treated group, the expression of Ki67 was significantly reduced in withanolides-treated groups. Ki67 positive cells/field of the withanolides 500 mg/kg group accounted for about 3.6% in the picture. There was also a statistically significant difference between the IMQ-treated group and the withanolides 500 mg/kg group.

### 2.3. Withanolides Inhibited the Infiltration of T Lymphocytes and Macrophages in Mice with IMQ-Induced Psoriasis

F4/80 was mainly expressed on the surface of macrophages, and stained brown particles in figure (Figure 5). More brown particles were observed in the IMQ-treated group compared with other groups, indicating a markedly increased infiltration of macrophages. F4/80 positive cells/field of IMQ-treated group accounted for about 5.5% in the picture. However, the expression of F4/80 was decreased in the withanolides 500 mg/kg group, and similar to that in the MTX-treated group. F4/80 positive cells/field of the withanolides 500 mg/kg group accounted for about 2.6% in the picture. It was demonstrated that withanolides could inhibited the infiltration of IMQ-induced macrophages.

The protein encoded by CD3E gene is the CD3-epsilon polypeptide, which together with CD3 and the T-cell receptor forms the T-cell receptor-CD3 complex. The complexes were stained brown particles though immunohistochemistry (Figure 6). Abundant brown particles were observed in the IMQ-treated group compared with other groups, indicating a markedly increased infiltration of T lymphocytes. CD3E positive cells/field of the IMQ-treated group accounted for about 7.7% in the picture. However, the expression of CD3E were decreased in the withanolides 500 mg/kg group, and lower than that in the MTX-treated group. F4/80 positive cells/field of the withanolides 500 mg/kg group accounted for about 3.6% in the picture. It was demonstrated that withanolides could inhibit the infiltration of IMQ-induced T lymphocytes

### 2.4. Withanolides Inhibited HaCat Cell Proliferation in a Dose-Dependent Manner

The impact of withanolides on the proliferation of HaCaT cells was analyzed via the CCK-8 assay. The optical density (OD) values increased in IMQ-stimulated HaCaT cells (see Figure 7). IMQ could stimulate HaCaT cell proliferation. However, the proliferation of HaCaT cells induced by IMQ was markedly inhibited after treatment with withanolides for 24 h. Withanolides could decrease OD value in a dose-dependent manner.

### 2.5. Withanolides Inhibited the Expression of Pro-Inflammatory Cytokines in IMQ-Stimulated Hacat Cells

We examined the efficacy of withanolides on the mRNA levels of some pro-inflammatory cytokines in IMQ-stimulated HaCaT cells. As shown in Figure 8, pre-treatment of HaCaT cells with withanolides could significantly inhibit IMQ-stimulated upregulations of IL-17, IL-6, TNF-α, IL-22 and CXCL1 mRNA.

### 2.6. Withanolides Could Inhibit Phosphorylations of STAT3, ERK1/2 and P38 in a Dose-Dependent Manner

We investigated the effect of withanolides on the STAT3, ERK1/2 and P38 pathways in IMQ-stimulated HaCaT cells. As shown in Figure 9, the relative protein expressions of p-ERK1/2 to total ERK1/2, p-STAT3 to total STAT3 and p-P38 to total P38 were decreased in IMQ-stimulated keratinocytes analyzed using western blot. Treatment with withanolides could effectively inhibit phosphorylations of STAT3, ERK1/2 and P38 in a dose-dependent manner. However, withanolides did not affect total levels of STAT3, ERK1/2 and P38. These results suggest that withanolides inhibited the activation of STAT3/ERK1/2/P38 signaling pathways in IMQ-stimulated HaCaT cells.

## 3. Discussion

In recent years, enormous efforts have been made in exploring the pathogenesis of psoriasis facilitating the development of anti-psoriatic drugs. The therapeutic use of natural product has been widely explored by human to treat psoriasis. Withanolides from the dried flowers of *Datura metel* L. with anti-psoriatic anti-inflammatory and anti-proliferative efficacies was discovered in this study. The therapeutic use of natural product has been widely explored for treatment of psoriasis. In this study, we investigated the anti-psoriasis mechanisms of action of withanolides in IMQ-induced in vivo and in vitro models.

We chose an IMQ-induced mouse model and an HaCaT cells model for evaluating the anti-psoriatic effects of withanolides. IMQ-induced psoriatic lesions in mice closely resemble plaque-type psoriasis in humans in terms of morphology and histological characteristics [23,25]. We successfully established an IMQ-induced psoriatic mouse model; in vivo experiment, the mice in IMQ-treated group had obvious signs of erythema, scaling, epidermal hyperplasia and inflammatory cell infiltration compared with the control group. Pre-treatment with withanolides could relieve these symptoms. Withanolides inhibited IMQ-induced high expression of Ki67 on the basal layer of epidermis. Withanolides also alleviated the infiltration of inflammatory cells, including T cells and macrophages. Therefore, we observed that the intragastric administration of withanolides alleviated psoriasis-like symptoms.

Accumulating studies have reported that IMQ can induce HaCaT cell hyperproliferation and mimic psoriasis-like cell model in vitro. IMQ-induced HaCaT cells have been extensively applied to the findings of anti-psoriatic drugs [26]. We also discovered that withanolides inhibited proliferation of IMQ-stimulated HaCaT cells in a concentration dependent manner via CCK-8 assay. IMQ has been reported to be an immune stimulator which can activate the Toll receptors (TLR7 and TLR8)-induced inflammatory reaction [23,27]. Withanolides inhibited IMQ-induced up-regulations of IL-17, IL-6, TNF-α, IL-22 and CXCL1 in HaCaT cells.

The activation of ERK1/2, STAT3 and P38 signaling pathways are associated with cell proliferation and differentiation [28,29]. We also found that the STAT3, P38 and ERK1/2 signaling pathways are involved in mediating the inflammatory reaction in withanolides-treated keratinocytes. It was found that withanolides blocked the expression of p-STAT3, p-ERK1/2, p-P38 in IMQ-stimulated keratinocytes, which demonstrated that inhibition of the STAT3, P38 and ERK1/2 signaling pathways might be related to the anti-psoriatic action of withanolides.

## 4. Materials and Methods

### 4.1. Materials

DMEM (High glucose) and FBS were purchased from Hyclone Bioscience (Logan, UT, USA). Cell Counting Kit-8 assay, protease and phosphatase inhibitors were obtained from APExBIO (Houston, TX, USA). PVDF membranes, blocking Buffer, SDS lysis buffer, 0.25% trypsin-EDTA solution, penicillin-streptomycin solution Protein Assay Kit were purchased from Beyotime Biotechnology Co. Ltd. (Shanghai, China). The STAT3, p-STAT3 (Tyr705), ERK1/2 and p-ERK1/2 (T202 + T204) primary antibodies were obtained from Abcam Co. Ltd. (Shanghai, China). CD3E, F4/80, Ki67, P38, p-P38 (Tyr323) and GAPDH primary antibodies were acquired from Bioss Biotechnology Co. Ltd. (Beijing, China). IRDye 680RD secondary antibodies were purchased from Rebiosci Biotechnology Co. Ltd. (Shanghai, China). Trizol reagent and PrimeScript™ Double Strand cDNA Synthesis Kit were acquired from Takara Bio Co. Ltd. (Dalian, China). 5% IMQ cream was obtained from Aldara (Loughborough, UK). IMQ powder were acquired from Sigma-Aldrich (St. Louis, MO, USA).

### 4.2. Preparation of Withanolides

The dried flowers of *Datura metel* L. were collected and purchased from Lingao County in China. The plant was verified by Prof. Zhenyue Wang (Department of Chinese Medicine Resources, Heilongjiang University of Chinese Medicine). Briefly, the dried flowers of *Datura metel* L. were extracted with 20 volumes of 70% EtOH after 24 h maceration at room temperature. A total of 70% EtOH extract was dissolved and suspended in 0.1% HCl–H_2_O. Then the suspension was filtered and exchanged through cation exchange resin (001 × 7). The concentrated filtrate was subjected to macro resin AB-8 crosslinked polystyrene and sequentially eluted with H_2_O, 50% EtOH, and 95% EtOH. 50% EtOH elution was concentrated, then was subject to macro resin D941 and sequentially eluted with H_2_O, 50% EtOH, and 95% EtOH. Withanolides was enriched in the section of H_2_O [30]. Withasteroids are a group of structurally diverse steroidal compounds with a C28 steroidal lactone skeleton, in which a characteristic feature is the presence of an α, β-unsaturated δ lactone ring in the side chain [20,21]. The UV absorption peaks of the withanolides are concentrated near the region of 225 nm detected by HPLC [30].

### 4.3. Psoriatic Model and Treatments

Eight-week-old female C57BL/6 mice were purchased from the Experimental Animal Centre of Harbin Medical University (Haerbin, China). All animal experimental protocols were approved and supervised by the Institutional Animal Care of Heilongjiang University of Chinese Medicine (Haerbin, China). They were supplied with food and water ad libitum. C57BL/6 mice were randomly divided into five groups (*n* = 10), including the control group (CG), methotrexate (MTX)-treated group, IMQ-treated group, withanolides 1 g/kg group, and withanolides 0.5 g/kg group. The back hair of all mice was shaved at a size of 2 × 2 cm^2^. In addition to the control group, other groups received 5% IMQ cream (62.5 mg/day) on their shaved backs for a consecutive 6 days to induce inflammation. Withanolides 0.5 g/kg and 0.25 g/kg groups were respectively treated with withanolides 0.5 g/kg and 0.25 g/kg in intragastric way once a day for 6 consecutive days. All mice were euthanized on day 7. Blood, skin and spleen samples of all mice were excised and collected for further biochemical and histological studies.

### 4.4. Scoring Severity of Skin Inflammation

A Psoriasis Area and Severity Index (PASI) was clinically developed to assess the severity of the inflammation of the psoriasis-like skin. The severity of the inflammation was daily determined by three clinical signs: erythema, infiltration, and scaling. The severity indexes of the three signs were independently evaluated on a score from 0 to 4: 0, none; 1, slight; 2, moderate; 3, severe; 4, very severe. The accumulative score (erythema + scaling + infiltration) served to demonstrate the severity of inflammation.

### 4.5. Histo-Pathological and Immunohistochemical Analysis

The back skin of all the mice was fixed in neutral buffered formalin and embedded in paraffin for 24 h. Tissue sections (3 μm) were sequentially cut, deparaffinized, and later stained with hematoxylin and eosin (H&E). The pathological changes were observed via an optical microscope (OLYMPUS BX60, Japan) with the help of well-trained pathologists. For immunohistochemistry staining, tissue sections were sequentially xylene-deparaffinized, ethanol-rehydrated, antigen-repaired and 3% hydrogen peroxide-quenched. The sections (3 μm) were stained with rabbit polyclonal anti-Ki67, rabbit polyclonal anti-F4/80 antibody, rabbit polyclonal anti-CD3E antibody. The staining was observed using a light microscope (OLYMPUS BX60, Tokyo, Japan). Positive areas of Ki67, F4/80, and CD3E in the epidermis were quantified with the digital medical image analysis system (Motic Med 6.0, Hong Kong, China).

### 4.6. HaCaT Cell Culture and Treatment

HaCaT cells were cultured in DMEM supplemented with 10% FBS, 100 U/mL penicillin, and 100 μg/mL streptomycin in a humidified atmosphere containing CO_2_ at 37 °C. The effects of withasteroids on cell viability were performed using a Cell Counting Kit-8 assay according to the manufacturer’s instructions. HaCaT cells were seeded in 96-well plates (3 × 10^4^ cell/well). Under 70–80% confluent conditions, they were stimulated with 1 μg/mL IMQ in the presence or absence of different concentration of withasteroids for 24 h. 10 μL CCK-8 was added to each well and incubated for 1 h. Later the optical density (OD) at 450 nm was measured using a microplate reader (BioTk, USA).

### 4.7. RT-PCR Assay

Total RNA was extracted from HaCaT cells using Trizol reagent. Total RNA was reverse transcribed and the complementary DNA was performed using PrimeScript™ Double Strand cDNA Synthesis Kit. The relative levels of following genes were quantified by Bio-Rad/CFX96 Touch™ Deep Well Real-Time PCR Detection (Biorad, CA, USA). PCR parameters were as follows: 95 °C for 10 min; 40 cycles at 95 °C for 15 s, and 60 °C for 60 s. The GAPDH gene was used as an internal control. The following genes were quantitatively analyzed according to the 2 -ΔΔCq method relative to GAPDH gene. The genes included: IL-17, IL-6, TNF-α, IL-22 and CXCL1.

### 4.8. Western Blot Analysis

HaCaT cells were harvested and lysed with an SDS lysis buffer supplemented with protease and phosphatase inhibitors. The concentration of proteins in the supernatant were quantified by Bradford Protein Assay. The proteins were denatured and subjected to 12% (*w*/*v*) SDS-PAGE. The electrophoresed proteins were transferred onto PVDF membranes. The membranes were blocked in blocking buffer for 1 h at room temperature, and then incubated with relevant primary antibodies overnight at 4 °C. Primary antibodies included: GAPDH, STAT3, p-STAT3(Tyr705), P38, p-P38(Tyr323), ERK1/2 and p-ERK1/2 (T202+204). The PVDF membranes were washed with PBST three times and then incubated with appropriate IRDye 680RD secondary antibodies for 1 h at 4 °C. The integrated protein bands were visualized by Odyssey CLx Imaging System (LICOR, NB, USA).

### 4.9. Statistical Analysis

The experimental data were analyzed using Graph Pad Prism 5 software and *p*-value < 0.05 was considered to be statistically significant. The experiment data were calculated as means ± SD. Data were performed using one-way analysis of variance (ANOVA) followed by Bonferroni’s Multiple Comparison tests.

## 5. Conclusions

Our results demonstrate that withanolides possess anti-inflammatory and anti-proliferative effects and significantly ameliorate the IMQ-induced psoriasis-like symptoms. Withanolides from *Datura metel* L. possess anti-inflammatory and anti-proliferative effects and might serve as promising components to fight against inflammation-type psoriasis.

## Figures and Tables

**Figure 1 molecules-24-02596-f001:**
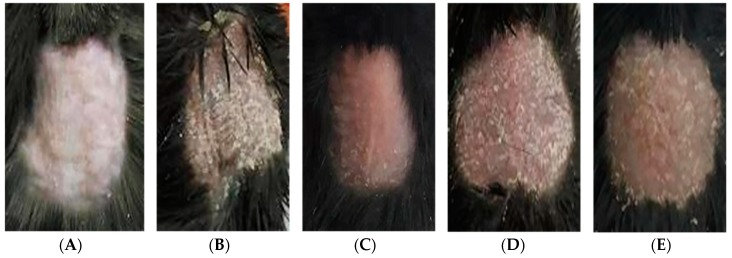
Morphological changes in skin lesions. (**A**) Control group. (**B**) Imiquimod (IMQ)-treated group. (**C**) Withanolides 500 mg/kg group. (**D**) Withanolides 250 mg/kg group. (**E**) Methotrexate (MTX)-treated group.

**Figure 2 molecules-24-02596-f002:**
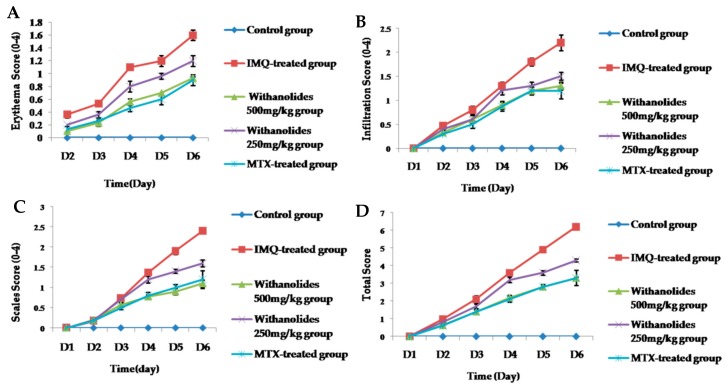
(**A**–**D**) The back skin of the five groups showed different grades of erythema, scales, infiltration and total score.

**Figure 3 molecules-24-02596-f003:**
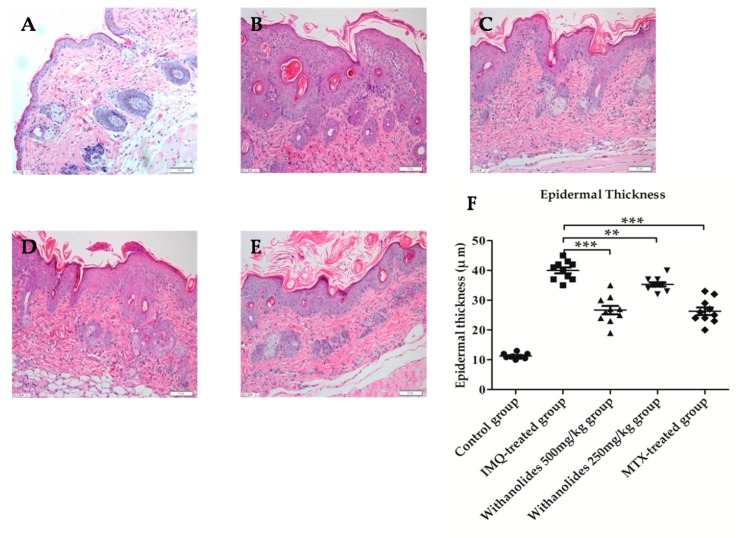
Effects of withanolides on histological changes in skin lesions of mice on day 6 of treatment (hematoxylin-eosin HE staining ×400. (**A**) Control group. (**B**) IMQ-treated group. (**C**) Withanolides 500 mg/kg group. (**D**) Withanolides 250 mg/kg group. (**E**) MTX-treated group. (**F**) Epidermal thickness in the dermis. Epidermal thickness in the dermis was measured in five randomly chosen fields (mean ± SD, *n* = 5). ** *p* < 0.01 vs. IMQ-treated group, *** *p* < 0.001 vs. IMQ-treated group.

**Figure 4 molecules-24-02596-f004:**
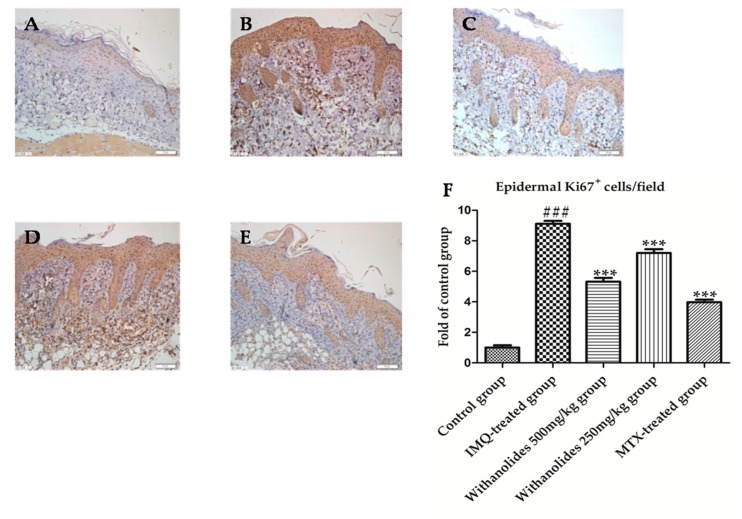
The psoriasis-like skin sections were strained by immunohistochemistry to evaluate Ki67 in five groups. (**A**) Control group. (**B**) IMQ-treated group. (**C**) Withanolides 500 mg/kg group. (**D**) Withanolides 250 mg/kg group. (**E**) MTX-treated group. (**F**) Ki67 positive cell/field (IF ×400). *** *p* < 0.001 vs. IMQ-treated group, ### *p* < 0.001 vs. control group.

**Figure 5 molecules-24-02596-f005:**
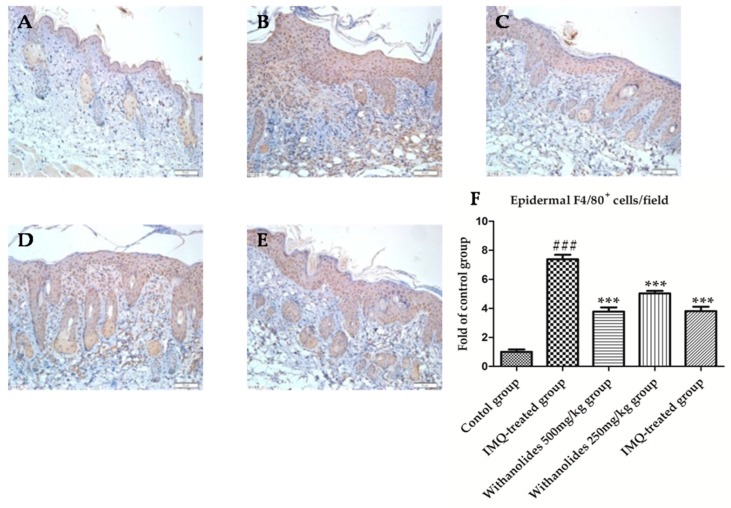
The psoriasis-like skin sections were strained by immunohistochemistry to evaluate F4/80 in five groups. (A) Control group. (**B**) IMQ-treated group. (**C**) Withanolides 500 mg/kg group. (**D**) Withanolides 250 mg/kg group. (**E**) MTX-treated group. (**F**) F4/80 positive cell/field (IF ×400). *** *p* < 0.001 vs. IMQ-treated group, ### *p* < 0.001 vs. control group.

**Figure 6 molecules-24-02596-f006:**
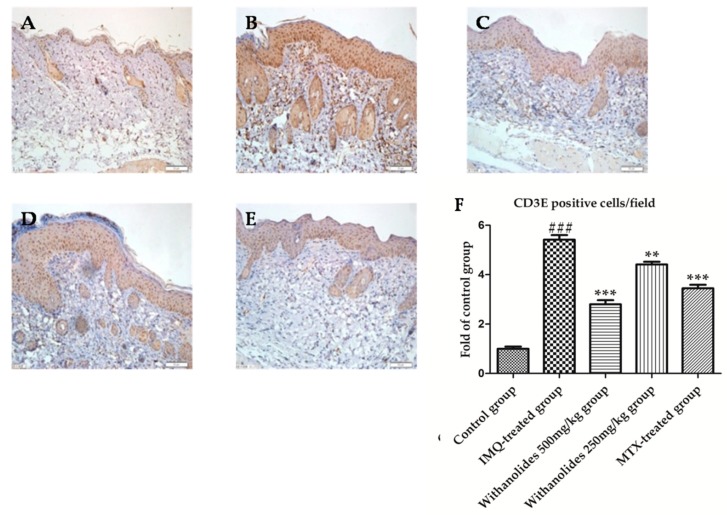
The psoriasis-like skin sections were strained by immunohistochemistry to evaluate CD3E in five groups. (**A**) Control group. (**B**) IMQ-treated group. (**C**) Withanolides 500 mg/kg group. (**D**) Withanolides 250 mg/kg group. (**E**) MTX-treated group. (**F**) CD3E positive cell/field (IF ×400). ** *p* < 0.01 vs. IMQ-treated group, *** *p* < 0.001 vs. IMQ-treated group, ### *p* < 0.001 vs. control group.

**Figure 7 molecules-24-02596-f007:**
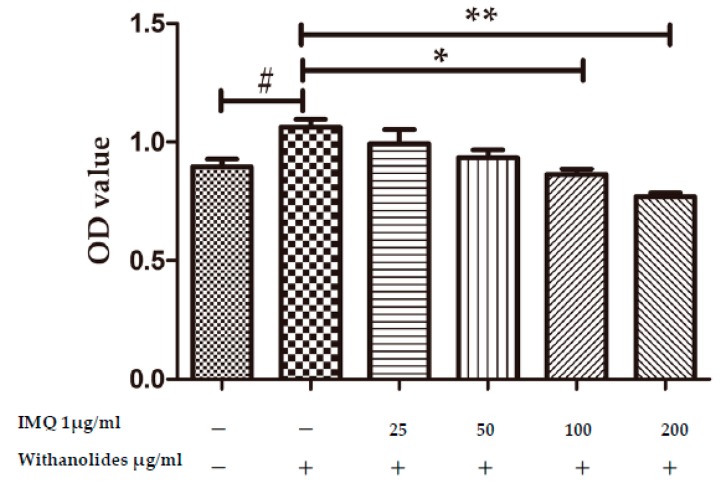
Effects of withanolides on HaCaT cell proliferation. HaCaT cells were treated with various doses of withanolides (0, 25, 50, 100 and 200 μg/mL) in the presence or absence of 1 μg/mL IMQ for 24 h. The optical density (OD) values were measured by the CCK-8 assay (mean ± SD, *n* = 3). # *p* < 0.05 vs. cell control group, * *p* < 0.05 vs. IMQ-treated group, ** *p* < 0.01 vs. IMQ-treated group.

**Figure 8 molecules-24-02596-f008:**
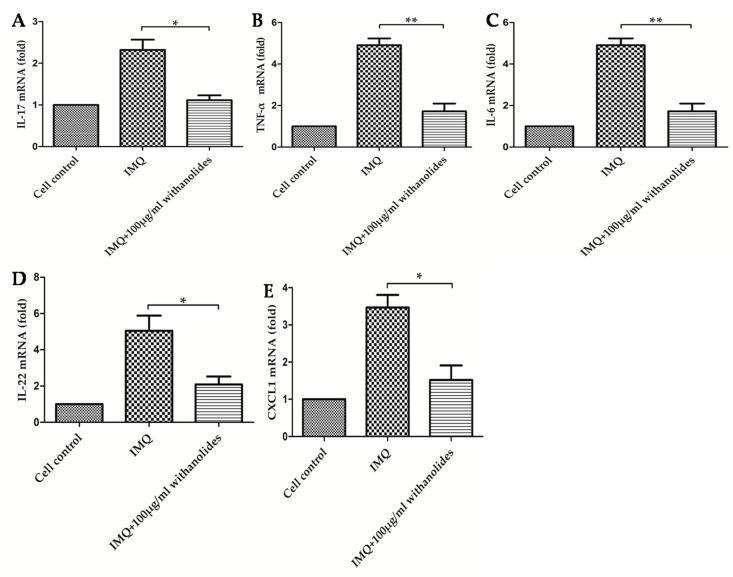
Effects of 100 μg/mL withanolides on relative mRNA levels of five pro-inflammatory cytokines in IMQ-stimulated HaCaT cells. (**A**) IL-17, (**B**) TNF-α, (**C**) IL-6, (**D**) IL-22, (**E**) CXCL1. (mean ± SD, *n* = 3) * *p* < 0.05 vs. IMQ group, ** *p* < 0.01 vs. IMQ group.

**Figure 9 molecules-24-02596-f009:**
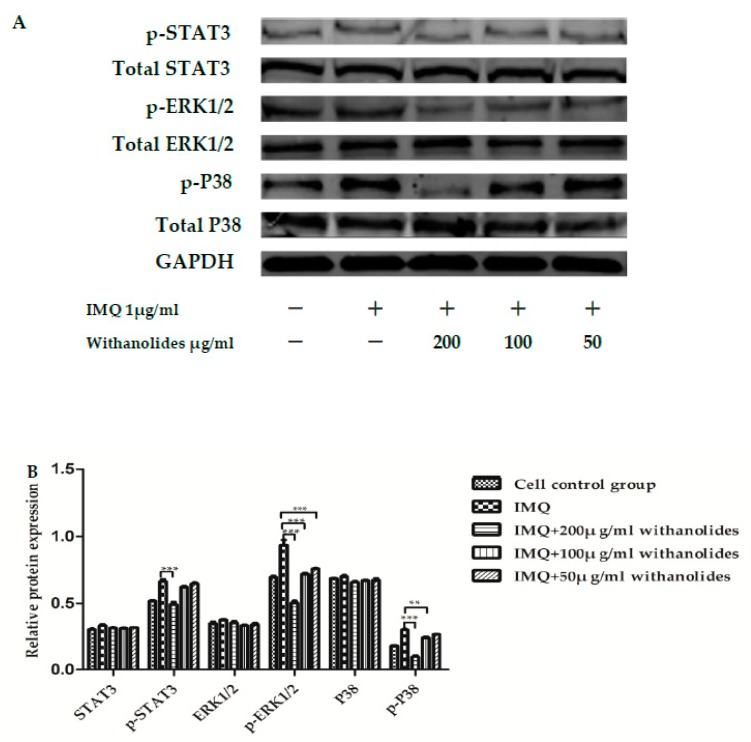
Effects of withanolides on relative protein expressions of six proteins in IMQ-stimulated HaCaT cells (**A**) Representative western blotting of keratinocytes with IMQ stimulation. (**B**) Representative protein levels in HaCaT cells. ** *p* < 0.01 *** *p* < 0.001 vs. IMQ group.

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
