# Peer review of "Withanolides, Extracted from Datura Metel L. Inhibit Keratinocyte Proliferation and Imiquimod-Induced Psoriasis-Like Dermatitis via the STAT3/P38/ERK1/2 Pathway"

_molecules, 2019, doi:10.3390/molecules24142596_

Round 1

Reviewer 1 Report

The manuscript „Withanolides, extracted from Datura metel L., …,“ is an interesting work providing new pre-clinical information for their potential use in the treatment of psoriasis in the future. Using keratinocytes and psoriasis-like mouse model they confirmed inhibitory effects of withanolides on immune responses after imiquimod induction. Several comments and recommendation should reduce formal mistakes existing and improve the quality of the MS.

1) page 2, lines 57-60: two last sentences should be moved to the Conclusions

2) page 3 – comments to Figures 3, 5 and 6 in the Results are missing!

3) Asterisks of significance (for p values) are too small for readers (Fig 4, 6, 7, etc.).

4) page 5 line 129 – the sentence is incomplete.

5) Figures in the MS are somewhat confusing. I recommend, for example, to move all parts of Figure 4 on a separate sheet, including the legend.

6) Why the authors sometimes used the capitals for withanolides (Fig 5-7, page 9 line 229) versus normal fonts (other parts of the MS)?

7) page 8 lines 170-3 please move to the Discussion

8) No Figure 8 mentioned in the Result text (page 8) is present

9) The authors used terms HaCaT and also Hacat cells in the MS, please select the better term from them.  

Author Response

Point 1: page 2, lines 57-60: two last sentences should be moved to the Conclusions

Response:Thank you very much for your comments. Page 2, lines 57-60: two last sentences have been moved to the Conclusions. Conclusions: Our results demonstrate that withanolides possess anti-inflammatory and anti-proliferative effects and significantly ameliorate the IMQ-induced psoriasis-like symptoms. Withanolides from Datura metel L. possess an anti-inflammatory and anti-proliferative effects and might serve as promising components to fight against inflammation-type psoriasis.They have been marked red in 316-319.

Point 2: page 3 – comments to Figures 3, 5 and 6 in the Results are missing!

Response: Thank you very much for your comments. The order of the pictures has been adjusted.  So, I have added the comments to Figures 3, 7 and 8 in the Results. They have been marked red in 84,149,161.

Point 3: Asterisks of significance (for p values) are too small for readers (Fig 4, 6, 7, etc.).

Response: Thank you very much for your comments. According to your suggestion I've increased the asterisks of significance (for p values) on all the graphs.

Point 4: page 5 line 129 – the sentence is incomplete.

Response: Thank you very much for your comments. The sentence is complete. It was demonstrated that withanolides could inhibited the infiltration of IMQ-induced T lymphocytes. It has been marked red in 135.

Point 5: Figures in the MS are somewhat confusing. I recommend, for example, to move all parts of Figure 4 on a separate sheet, including the legend.

Response: Thank you very much for your comments.  I have divided figure 4 into figure 4, 5 and 6, including the legend.

Point 6: Why the authors sometimes used the capitals for withanolides (Fig 5-7, page 9 line 229) versus normal fonts (other parts of the MS)?

Response: Thank you very much for your comments.  I have changed the capitals for withanolides (Fig 7-9, page 11 line 236) to normal fonts. They have been marked red in

154, 165, 175, 236.

 Point 7: page 8 lines 170-3 please move to the Discussion

Response: Thank you very much for your comments. I have moved the sentences to the Discussion. “The activation of ERK1/2, STAT3 and P38 signaling pathways are associated with cell proliferation and differentiation .” They have been marked red in 213, 214.

Point 8: No Figure 8 mentioned in the Result text (page 8) is present

Response: Thank you very much for your comments. I am really sorry. I marked the number of pictures incorrectly. It should be picture 9. I have corrected it. It has been marked red in 180.

Point 9: The authors used terms HaCaT and also Hacat cells in the MS, please select the better term from them.

Response:  Thank you very much for your comments. I am really sorry. I've changed all the Hacat to HaCaT. They have been marked red in the manuscript.

Reviewer 2 Report

The manuscript “Withanolides, extracted from Datura metel L. inhibit  keratinocyte proliferation and imiquimod- induced  psoriasis-like dermatitis via the STAT3/P38/ERK1/2  pathway” presented by Tingting Li, Zheng Wei, Yanping Sun, Qiuhong Wang, Haixue Kuang is very interesting work dedicated to the healing effects of Datura flowers on psoriasis. Psoriasis is affecting about 4% of European population and is extremely devastating condition with many complications and the confirmation that some natural compounds can reduce psoriasis effects is  very important.

The work is well planned and accurately done. Manuscript is written in good English.

This reviewer will only ask authors to made very small changes to the manuscript and after will recommend it to be accepted:

1.      Please, add to the discussion the explanation why you prefer intra gastric application of withanolides and not the injection or just the topic application? Many steroids have very limited intestinal adsorption (Fleisher et al, 1986  J Pharm Sci. 1986 Oct;75(10):934-9). Is it a standard application for traditional medicine?

2.      Can you please add more references on your previous characterization of withanolides in Datura extracts to the method Section. I know that you previously characterized withanolides in extracts with  NMR and mass spectrometry, and this must be clearly stated in Methods.

There is some small misspelling in text, like on Line 54 (minic instead of mimic). But it does not affect understanding of you text.

Author Response

Point 1:    Please, add to the discussion the explanation why you prefer intra gastric application of withanolides and not the injection or just the topic application? Many steroids have very limited intestinal adsorption (Fleisher et al, 1986  J Pharm Sci. 1986 Oct;75(10):934-9). Is it a standard application for traditional medicine?

Response: Thank you very much for your comments. Firstly, gastric application of withanolides is safe and convenient. The above experimental results confirmed that gastric application of withanolides extracted from Datura metel L. achieved good  therapeutic effects. Withasteroids are a group of structurally diverse steroidal compounds with a C28 steroidal lactone skeleton, in which a characteristic feature is the presence of an α, β-unsaturated δ lactone ring in the side chain. The chemical components and structures of traditional Chinese medicine are complex. Different traditional Chinese medicines should adopt the most appropriate mode of administration through the pre-experimental verification.

Point 2:  Can you please add more references on your previous characterization of withanolides in Datura extracts to the method Section. I know that you previously characterized withanolides in extracts with  NMR and mass spectrometry, and this must be clearly stated in Methods.

Response: Thank you very much for your comments. Withasteroids are a group of structurally diverse steroidal compounds with a C28 steroidal lactone skeleton, in which a characteristic feature is the presence of an α, β-unsaturated δ lactone ring in the side chain. The UV absorption peaks of the withanolides are concentrated near the  region of 225nm detected by HPLC. The purity of the Withasteroids reaches 94% analyze using HPLC, NMR and MS. I have added some references on characterization of withanolides in Datura extracts to the method Section. They have been marked red in 264-250.

Point 3: There is some small misspelling in text, like on Line 54 (minic instead of mimic). But it does not affect understanding of you text.

Response: Thank you very much for your comments. I've changed minic to mimic. It has been marked red in 54.

Reviewer 3 Report

In this article authors investigated the effect of withanolides extracted from Datura metel L. plant flowers on in vitro keratinocyte proliferation and in vivo induced psoriasis-like dermatitis using C57BL/6 mouse model.

There are some additional points to be resolved:

What authors did, that they simultaneously induce psoriasis-like skin manifestations and introduce withanolides extract. It would also be relevant that withanolides extract is administered after the end of exposure to IMQ and to see how PASI is improved over time. Could authors comment on that?   

Question regarding withanolides extract, do you know which compounds are present in this extract, major/minor? Is it possible to link these effects to certain compound or compounds? Could there be any additive or synergistic effect? If this can be achieved with single compound it might be a prospect for pharmaceutical application.

Typfellers: Line 14, 54, 200, 250.

A number of sentences are grammatically incorrect, for example line 248  

4.9. Statistical analysis

Author stated that “The means ± S.E.M. of experiment data were calculated at least three experiments.”

If grammar is disregarded, this is rather confusing sentence, it reads like animals went through same experiment 3 time? So, what is the meaning of this sentence? If that is the case how the data were treated statistically? Looking at figure 3F, I see only 10 points. Could you please clarify this and adopt statistics if necessary or give explanation in the figure captions, depending on the answer. Figure captions should be self-explained without the text and some are not. In figure 4 you have used fold increase, what are the absolute values (should be given in the text).

Authors used ANOVA with Bonferroni’s post test, how did you check for data distribution?  

4.3. Psoriatic model and treatments

Authors used as designation 1g/kg and 0.5g/kg for withanolides treated groups and further they stated that these groups were treated with 0.5g/kg and 0.25g/kg withanolides extracts, respectively. Where 1g/kg and 0.5g/kg groups come from? What was the dose after all, which one is correct? I would suggest to explain this and put into text or to change the designation to be correct.

4.4. Scoring severity of skin inflammation

Psoriasis Area and Severity Index (PASI) was clinically developed to assess the severity of the inflammation of the psoriasis-like skin.”

PASI was developed some time ago, so you use it in this study, but you did not develop scoring system!?  

Author Response

Point 1: What authors did, that they simultaneously induce psoriasis-like skin manifestations and introduce withanolides extract. It would also be relevant that withanolides extract is administered after the end of exposure to IMQ and to see how PASI is improved over time. Could authors comment on that?

Response: Thank you very much for your comments. PASI scores of IMQ-treated group exhibited  the highest compared with other groups over time. However, PASI scores of withanolides-treated groups exhibited obviously lower than IMQ-treated group. Withanolides could significantly inhibit IMQ-stimulated degree of up-regulation of PASI scores over time. The decrease in degree of up-regulation of PASI suggested that withanolides had anti-psoriatic efficacy compared with the IMQ-treated group.

Point 2: Question regarding withanolides extract, do you know which compounds are present in this extract, major/minor? Is it possible to link these effects to certain compound or compounds? Could there be any additive or synergistic effect? If this can be achieved with single compound it might be a prospect for pharmaceutical application.

Response: Thank you very much for your comments. Thank you very much for your suggestions. The following experiments will confirm which compounds are present in this withanolides and the relationship of these effects and certain compound or compounds.

Point 3: A number of sentences are grammatically incorrect, for example line 248

Response: Thank you very much for your comments. The sentence has been corrected. “The back hair of all mice was shaved at a size of 2 × 2 cm2 .” It has been marked red in 258.

Point 4: Author stated that “The means ± S.E.M. of experiment data were calculated at least three experiments.”

If grammar is disregarded, this is rather confusing sentence, it reads like animals went through same experiment 3 time? So, what is the meaning of this sentence? If that is the case how the data were treated statistically? Looking at figure 3F, I see only 10 points. Could you please clarify this and adopt statistics if necessary or give explanation in the figure captions, depending on the answer. Figure captions should be self-explained without the text and some are not. In figure 4 you have used fold increase, what are the absolute values (should be given in the text).

Response: Thank you very much for your comments. The sentence has been changed to “The experiment data were calculated as means ± SD.” I have added the absolute values in figure 4, 5 and 6. They have been marked red in 107, 111, 123, 125, 131, 133, 134.

Point 5: Authors used ANOVA with Bonferroni’s post test, how did you check for data distribution?  

Response: Thank you very much for your comments. The sentence has been changed to“Data were performed using one-way analysis of variance (ANOVA) followed by Bonferroni's Multiple Comparison tests. ” Data distribution were analyzed using Graph Pad Prism 5 software. P-value < 0.05 was considered to be statistically significant. 

Point 6: Psoriatic model and treatments

Authors used as designation 1g/kg and 0.5g/kg for withanolides treated groups and further they stated that these groups were treated with 0.5g/kg and 0.25g/kg withanolides extracts, respectively. Where 1g/kg and 0.5g/kg groups come from? What was the dose after all, which one is correct? I would suggest to explain this and put into text or to change the designation to be correct.

Response: Thank you very much for your comments. I'm so sorry. 0.5g/kg and 0.25g/kg withanolides extracts are correct. I have corrected them. They have been marked red in 258.

Point 7: Scoring severity of skin inflammation

“Psoriasis Area and Severity Index (PASI) was clinically developed to assess the severity of the inflammation of the psoriasis-like skin.”

PASI was developed some time ago, so you use it in this study, but you did not develop scoring system!?  

Response: Thank you very much for your comments. I did not develop scoring system. PASI was  scored with the help of well-trained pathologists, according to the standard of PASI.
